# Hybrid Regularization Methods Achieve Near-Optimal Regularization in Random Feature Models

## Abstract

We demonstrate the potential of hybrid regularization methods to automatically and efficiently regularize the training of random feature models to generalize well on unseen data. Hybrid methods automatically combine the strengths of early stopping and weight decay while avoiding their respective weaknesses. By iteratively projecting the original learning problem onto a lower-dimensional subspace, they provide an efficient way to choose the weight decay hyperparameter. In our work, the weight decay hyperparameter is automatically selected by generalized cross-validation (GCV), which performs leave-one-out cross-validation simultaneously in a single training run and without the need for a dedicated validation dataset. As a demonstration, we use the random feature model to generate well- and ill-posed training problems arising from ~~image classification~~four image classification and regression datasets. Our results show that hybrid regularization leads to near-optimal regularization in all problems. In particular, it is competitive with optimally tuned classical regularization methods. While hybrid regularization methods are popular in many large-scale inverse problems, their potential in machine learning is under-appreciated, and our findings motivate their wider use. We provide our MATLAB codes for implementing the numerical experiments in this paper at `Github-link-made-anonymous-for-reviewing`.

## 1 Introduction

We examine the ability of hybrid regularization methods to automatically and efficiently train machine learning models that generalize, that is, to perform well on data that have not been used in training. Hybrid regularization methods are a class of efficient regularization methods to tackle ill-posed linear inverse problems. They synergistically combine the two most common and successful regularization schemes: early stopping (Yao et al., 2007), which is also known as iterative regularization (Engl et al., 1996, Chapter 6) in inverse problems, and weight decay (Goodfellow et al., 2016, Chapter 7)~~(also called iterative regularization (Engl et al., 1996, Chapter 6) and~~, which is also called ridge penalty in statistics/machine learning (Kobak et al., 2020) or Tikhonov regularization (Engl et al., 1996, Chapter 5) in inverse problems ~~, respectively).~~ In particular, hybrid regularization combines the respective strengths of the two classical regularization schemes while circumventing their drawbacks. Developing hybrid methods has been a fruitful and important direction in inverse problems recently (Chung & Palmer, 2015; Gazzola et al., 2015; Chung & Saibaba, 2017; Chung & Gazzola, 2024). Their effectiveness has also been documented in other fields, including various imaging problems (Chung et al., 2008; Chung & Palmer, 2015; Chung & Saibaba, 2017).

In this paper, we consider a computationally efficient hybrid method (Chung et al., 2008) that automatically tunes its hyperparameters and leads to trained models that generalize well on different problems. To be specific, we use the hybrid method implemented in Gazzola et al. (2018), which performs a few iterations of the numerically stable Krylov subspace method LSQR (Paige & Saunders, 1982a;b) and adaptively selects the weight decay hyperparameter at each iteration using generalized cross-validation (GCV) (Golub et al., 1979). Notably, the scheme does not require a dedicated validation dataset.

To test the effectiveness of hybrid regularization, we consider the training of random feature models (RFM). RFM are machine learning models that express the relationship between given input and output

features as the concatenation of a random feature extractor and a linear model whose weights are optimized. It has been observed that the RFM's generalization gap decreases as the number of random features grows, sharply spikes when it reaches the number of training data, and decays as the number of random features is increased further. This is called the double descent phenomenon in existing litera-ture ~~(Belkin et al., 2019; 2020; Hastie et al., 2022; Advani et al., 2020; Ma et al., 2020; Mei & Montanari, 2022)~~ (Belkin et al., 2019; 2020; Hastie et al., 2022; Advani et al., 2020; Ma et al., 2020; Mei & Montanari, 2022; Li et al., 2020; K .

To understand why hybrid regularization is adequate for RFMs, we provide an inverse problems perspective and relate the spike in the generalization gap to the ill-posedness of the training problem. Specifically, the training problem is the most ill-posed when the number of random features is equal to the num-ber of training data. This allows us to create training problems with different levels of ill-posedness by simply varying the number of random features in the experiments. Moreover, one can tackle the ill-posedness and improve generalization by early stopping and weight decay regularization - provided an effective choice of ~~parameters; we therefore extend similar arguments made~~ hyperparameters. Indeed, these effects have been studied in extensive literature; see, e.g., ~~by Advani et al. (2020); Ma et al. (2020).~~ Raskutti et al. (2014); Ali et al. (2019); Advani et al. (2020); Shen et al. (2022); Sonthalia et al. (2024) for early stopping, Hastie et al. (2022); Wu & Xu (2020); Kobak et al. (2020); Nakkiran et al. (2021); Sonthalia et al. (2023) for weight decay, and Bishop (1995); Dhifallah & Lu (2021) for general Tikhonov regularization. While these existing works provide insights into and have similarities with the hybrid approach here, the combination of low-rank projections and generalized cross-validation to achieve automatic and efficient hyperparameter tuning sets the present work apart. Unlike most existing works, the hybrid approach neither requires a dedicated validation set nor a full matrix factorization because the regularization hyperparameter and stopping iteration can be selected using the generalized cross-validation (GCV) function in a low-dimensional subspace. In addition, while some existing works assume certain data distributions, hybrid methods can flexibly pair with agnostic hyperparameter selection schemes, e.g., GCV.

While the success of hybrid regularization in inverse problems has been documented, their application to ma-chine learning has not been extensively studied. Our work is also motivated by Newman et al. (2022), which shows the benefits of hybrid regularization to adaptively choose learning rate and weight decay parameters for the weights associated with the last layer of deep networks. To gain more insight into the potential of hybrid methods to improve generalization, we perform extensive experiments using ~~RFM, and the MNIST and CIFAR-10~~ two different variants of RFM and four commonly used datasets. We compare the effective-ness of early stopping, weight decay, and hybrid regularization for training problems with different levels of ill-posedness. In our experiments, hybrid regularization leads to trained models whose generalization is comparable with classical regularization approaches even when their hyperparameters are tuned to minimize the test loss, a procedure that is to be avoided in realistic applications. This suggests the potential of hybrid methods as a generic algorithm to train machine learning models reliably. To enable reproducibility, we provide our codes used to perform the experiments at `Github-link-made-anonymous-for-reviewing`.

The remainder of the paper is organized as follows. In Section 2, we set up the problem and review the classical early stopping and weight decay regularization methods. In Section 3, we describe a hybrid regu-larization scheme and outline its advantages. In Section 4, we describe the setup of RFM training and relate the deterioration in its generalization gap to its ill-posedness. In Section 5, we demonstrate that hybrid regularization can effectively and automatically train RFM that generalize well with extensive experiments. ~~Finally, we~~ We conclude the paper in Section 6. Additional experimental results are shown in Appendix A.

## 2 Regularization by Early Stopping and Weight Decay

In this section, we layout the setup of the training problem and review the definition of ill-posed problems. We then study the regularizing effects of two classical schemes: early stopping and weight decay. Weight decay is also called ridge penalty in statistics/machine learning. In the inverse problems literature, they are better known as iterative regularization and Tikhonov regularization, respectively.

**Problem Setup** Given a matrix of input features $\mathbf{A} \in \mathbb{R}^{m \times n}$ and the corresponding output $\mathbf{b} \in \mathbb{R}^m$, where the $m$ examples are stored row-wise, and $n$ is the dimension of the input features. We seek to identify a linear transformation $\mathbf{x} \in \mathbb{R}^m$ such that $\mathbf{Ax} \approx \mathbf{b}$. To this end, the solution to an unregularized problem can be obtained by solving a linear least squares problem

$$\mathbf{x}_{\text{LS}}^* \in \arg\min_{\mathbf{x} \in \mathbb{R}^n} \frac{1}{2m} \|\mathbf{Ax} - \mathbf{b}\|_2^2. \tag{1}$$

In machine learning, the actual goal in the training problem (1) is not necessarily to solve (1) optimally but to obtain a linear transformation that generalizes beyond the training data. To gauge the generalization of the model defined by $\mathbf{x}_{\text{LS}}^*$, consider the test data set given by $\mathbf{A}_{\text{test}} \in \mathbb{R}^{m_{\text{test}} \times n}$ and $\mathbf{b}_{\text{test}} \in \mathbb{R}^{m_{\text{test}}}$. Then, the model defined by $\mathbf{x}_{\text{LS}}^*$ generalizes well if the generalization gap (Bengio et al., 2017, Figure 5.3), (He et al., 2022; Yang et al., 2023)

$$\frac{1}{2m_{\text{test}}} \|\mathbf{A}_{\text{test}} \mathbf{x}_{\text{LS}}^* - \mathbf{b}_{\text{test}}\|_2^2 - \frac{1}{2m} \|\mathbf{A} \mathbf{x}_{\text{LS}}^* - \mathbf{b}\|_2^2$$

is sufficiently small.

**Ill-posedness and Regularization** Regularization techniques are commonly used to improve the generalization of machine learning (see, e.g., (Goodfellow et al., 2016, Chapter 7)) and to enhance the solution of ill-posed inverse problems (see, e.g., Engl et al. (1996); Hansen (1998; 2010)). Despite differences in notation and naming, the fundamental ideas in both domains are similar.

To illustrate ill-posedness and the effects of regularization, we consider the singular value decomposition (SVD) of $\mathbf{A} = \mathbf{U\Sigma V}^\top$ in (1). Here, $\mathbf{U} = [\mathbf{u}_1, \mathbf{u}_2, ..., \mathbf{u}_n]$ and $\mathbf{V} = [\mathbf{v}_1, \mathbf{v}_2, ..., \mathbf{v}_m]$ are orthogonal matrices, and $\mathbf{\Sigma} \in \mathbb{R}^{n \times m}$ contains the singular values $\{\sigma_j\}_{j=1}^{\min(m,n)}$ in descending order on its diagonal and is zero otherwise. Let $r$ be the rank of $\mathbf{A}$, that is, the last index such that $\sigma_r > 0$.

When the singular values decay to zero smoothly, it is common to call problem (1) ill-posed and a large generalization gap is expected for some test data. To see this, consider the minimum norm solution of (1), which using the SVD can be written explicitly as

$$\mathbf{x}_{\text{LS}}^* = \sum_{j=1}^r \frac{\mathbf{u}_j^\top \mathbf{b}}{\sigma_j} \mathbf{v}_j. \tag{2}$$

This formulation shows that the contribution of $\mathbf{v}_j$ to $\mathbf{x}_{\text{LS}}^*$ is scaled by the ratio between $\mathbf{u}_j^\top \mathbf{b}$ and $\sigma_j$. If this ratio is large in magnitude then the solution is highly sensitive to perturbations of the data $\mathbf{b}$ along the direction $\mathbf{u}_j$ and the corresponding singular vector $\mathbf{v}_j$ gets amplified in the solution. Also, $\sigma_j$ quantifies the importance of the feature $\mathbf{u}_j$ in the data matrix $\mathbf{A}$. Therefore, intuitively one wishes that the ratio between $|\mathbf{u}_j^\top \mathbf{b}|$ ~~to decay~~ and $\sigma_j$ ~~to plunge to 0 sharply~~ stays bounded as $j$ grows. Otherwise, the weakly present features in $\mathbf{A}$ will be amplified and dominate the solution. If the resulting model is dominated by less relevant features, it is expected to have a large generalization gap for some test data $\mathbf{A}_{\text{test}}$ and $\mathbf{b}_{\text{test}}$. In other words, this observation suggests that for ill-posed problems, the generalization gap depends on the decay of $\sigma_j$ and $|\mathbf{u}_j^\top \mathbf{b}|$. See Figure 3(c)-(e) for an example of how the quantities in (2) behave for well- and ill-posed problems.

Using the SVD of the data matrix $\mathbf{A} = \mathbf{U\Sigma V}^\top$, we can write and analyze most regularization schemes for (1) using their corresponding filter factors $\phi_j$ that control the influence of the $j$th term in (2). To be precise, the regularized solutions can be written as

$$\mathbf{x}_{\text{reg}} = \sum_{j=1}^r \phi_j \frac{\mathbf{u}_j^\top \mathbf{b}}{\sigma_j} \mathbf{v}_j. \tag{3}$$

Here $\phi_j$ depends on the regularization hyperparameter, which we will specify in the following discussion. A simple example is the truncated SVD, in which terms associated with small singular values are ignored by

using the filter factors

$$\phi_{\text{TSVD},j}(\tau) = \begin{cases} 1, & \sigma_j > \tau, \\ 0, & \sigma_j \leq \tau, \end{cases} \tag{4}$$

where the choice of $\tau \geq 0$ is crucial to trade off the reduction of the training loss and the regularity of the solution, which is needed to generalize.

**Early Stopping**  A common observation when (1) is solved using iterative methods is an initially sharp decay of both the training and test losses followed by a widening of the generalization gap in later iterations. This behavior, also known as semiconvergence, typically arises when the iterative method converges quicker on the subspace spanned by the singular vectors associated with large singular values than on those associated with small singular values. A straightforward and popular way to regularize the problem then is to stop the iteration early.

As a simple example to show the regularizing effect of early stopping, we consider the gradient flow (GF) applied to (1), which reads

$$\partial_t \mathbf{x}_{\text{GF}}(t) = -\frac{1}{n} \mathbf{A}^\top (\mathbf{A} \mathbf{x}_{\text{GF}}(t) - \mathbf{b}), \quad \mathbf{x}_{\text{GF}}(0) = \mathbf{0}. \tag{5}$$

When the SVD of the feature matrix is available, $\mathbf{x}_{\text{GF}}(t)$ can be computed via (3) using the filter factors (Ma et al., 2020)

$$\phi_{\text{GF},j}(t) = 1 - e^{-\sigma_j^2 t/(mn)}. \tag{6}$$

From this observation, we can see that as $t$ grows, the filter factors converge to one, and $\mathbf{x}_{\text{GF}}(t)$ converges to the solution of the unregularized problem (1). Furthermore, we see that for any fixed time, the filter factors decay as $j$ grows, which reduces the sensitivity to perturbations of the data $\mathbf{b}$ along the directions associated with small singular values.

In the top row of Figure 1, we show numerical results for the early stopping applied to ~~two~~ four test problems. The qualitative behavior is comparable for ~~both~~ all datasets: Initially, both training and test losses decay with no noticeable gap but at later times, the test losses increase dramatically. A difference ~~between the two~~ across the datasets is that the optimal stopping time (i.e., the time with the smallest test loss) ~~differs by about two~~ can differ by about three orders of magnitude. Hence, the stopping time is the key hyperparameter that needs to be chosen judiciously and depends on the problem. Determining an effective stopping time is even more difficult in realistic applications, when this decision must not be based on the test dataset. This is in stark contrast with the hybrid scheme, where neither semiconvergence nor sensitivity to stopping time is observed; see Figure 2.

The cross-validation of early stopping is straightforward to perform, for example, one can use a part of the training data for validation and stop the training process when the validation error is minimized. However, such an approach reduces the number of training data. The regularization properties and their analysis also depend heavily on the underlying iterative method. Moreover, early stopping generally prefers slowly converging schemes to have a broader range of optimal stopping points.

**Weight Decay**  The idea in weight decay is to incorporate an extra term $\frac{\alpha^2}{2}\|\mathbf{x}\|_2^2$ into the objective in (1), which results in

$$\min_{\mathbf{x} \in \mathbb{R}^n} \frac{1}{2m} \|\mathbf{A}\mathbf{x} - \mathbf{b}\|_2^2 + \frac{\alpha^2}{2}\|\mathbf{x}\|_2^2. \tag{7}$$

Here the hyperparameter $\alpha \geq 0$ trades off the minimization of the loss and the regularity of the solution. It is noteworthy that there are other options for choosing the regularization term (Hansen, 1998, Section 4.3). In this work, we focus on the squared Euclidean norm for simplicity of illustration. An advantage compared to early stopping is that any iterative or direct method used to solve (7) will ultimately provide the same solution.

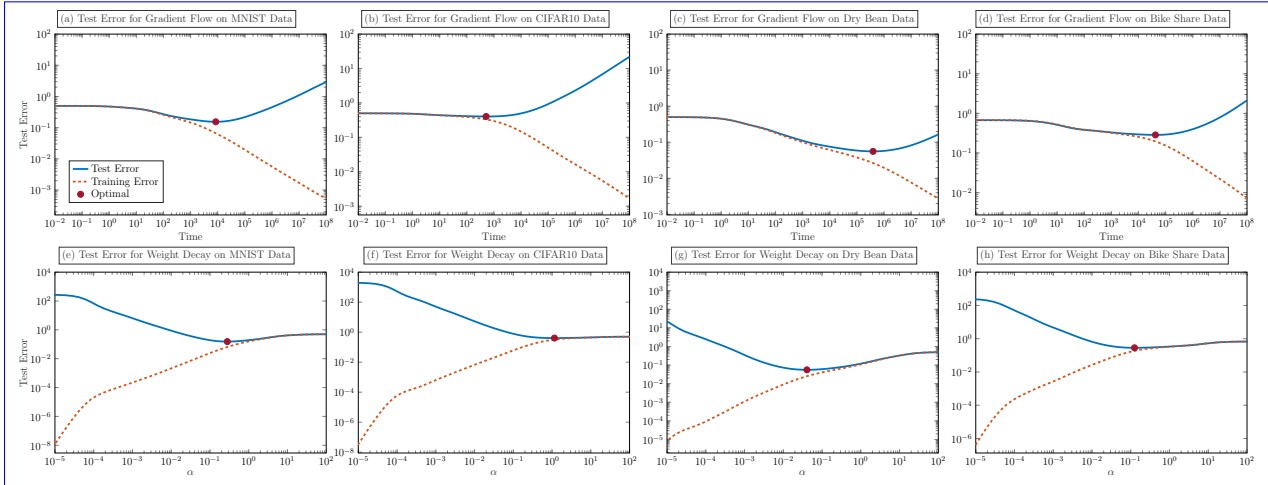

Figure 1: (a)-(~~b~~d): The results of applying gradient flow (GF) (5) to ~~two~~ four training problems at different time $t$. (~~e~~e)-(~~d~~h): The results obtained from weight decay (7) with different hyperparameter $\alpha$. The optimal stopping time/regularization hyperparameter is highlighted. We can see that both methods exhibit a semi-convergence behavior. In particular, their generalizability depends on the choice of hyperparameters, which varies from problem to problem and has to be made judiciously. Determining the optimal hyperparameters requires access to the test data. Yet, the access to test data is prohibited during training.

Using the SVD of $\mathbf{A}$ we can see that the weight decay solution $\mathbf{x}_{\text{WD}}(\alpha)$ can be computed using (3) and the filter factors are given by

$$\phi_{\text{WD},j}(\alpha) = \frac{\sigma_j^2}{\sigma_j^2 + m\alpha^2}, \tag{8}$$

which are called the Tikhonov filter factors (Hansen, 1998). We can see that when $\alpha$ is chosen relatively small, the filter factors associated with large singular values remain almost unaffected while those corresponding to small singular values may be close to zero. Thus, similar to TSVD and early stopping, weight decay can reduce the sensitivity of $\mathbf{x}_{\text{WD}}$ to perturbations of the data along the directions associated with small singular values. The Tikhonov filter gives a simple representation of the solution of weight decay. This renders its analysis straightforward. Moreover, the same solution is obtained regardless of the choice of solvers. Thus, in contrast to early stopping, the most efficient scheme can be used.

We investigate the impact of choosing $\alpha$ on the generalization in a numerical experiment for ~~the MNIST and CIFAR10 example; see Figure 1(c)-(d)~~ four datasets; see the bottom row of Figure 1. The qualitative behavior is comparable for ~~both~~ all datasets: as $\alpha$ increases the training error increases monotonically, while the test error first decays and then finally grows. Due to this semiconvergence, a careful choice of $\alpha$ can improve generalization; we visualize the hyperparameter $\alpha$ that yields the lowest test loss with red dots. A key difference ~~between~~ across the examples is that the optimal values of $\alpha$ can differ by about one order of magnitude, which highlights its problem-dependence. As in early stopping, we re-iterate that the test data must not be used to select the optimal value of $\alpha$. The optimal choice of the hyperparameter $\alpha$ requires cross-validation (Kohavi et al., 1995) in which the problem has to be solved many times.

## 3  Hybrid Regularization: The Best of Both Worlds

In this section, we describe a hybrid regularization scheme. We then highlight its advantages over classical regularization methods to motivate its usage for training machine learning models.

**Hybrid Regularization**  Hybrid regularization methods belong to the most effective solvers for ill-posed inverse problems. The key idea in hybrid methods is that they synergistically combine the respective advantages of early stopping and weight decay while avoiding their disadvantages. In this work, we consider

the technique `IRhybrid_lsqr` from the open source MATLAB package (Gazzola et al., 2018). This hybrid method employs LSQR (Paige & Saunders, 1982a;b) that at each iteration projects the regularized least-squares problem (7) onto a small-dimensional subspace and adaptively selects the weight decay parameter using generalized cross-validation (GCV) (Golub et al., 1979). Notably, the resulting hybrid method does not require any hyperparameter tuning or a dedicated validation dataset.

**LSQR Algorithm** LSQR (Paige & Saunders, 1982a;b) is an iterative method for solving (regularized) least squares problems. With comparable computational costs per iteration, the numerical stability and convergence of LSQR generally are superior to gradient descent, particularly for ill-posed problems. Moreover, LSQR is suitable for both dense and sparse large-scale problems because it does not require building the data matrix explicitly, but only the function handle to perform its matrix-vector products Chung et al. (2008); Chung & Gazzola (2024). The $k$th iteration of LSQR solves the projection of (7) onto the $k$-dimensional Krylov subspace $\mathcal{K}_k = \mathrm{span}\{\mathbf{A}^\top\mathbf{b}, (\mathbf{A}^\top\mathbf{A})\mathbf{A}^\top\mathbf{b}, \ldots, (\mathbf{A}^\top\mathbf{A})^{k-1}\mathbf{A}^\top\mathbf{b}\}$. This projection is obtained using Golub-Kahan bidiagonalization (Golub & Kahan, 1965) of the data matrix $\mathbf{A}$ with the initial vector $\mathbf{b}$, which reads

$$\mathbf{A}^\top\mathbf{Q}_k = \mathbf{P}_k\mathbf{L}_k^\top + \gamma_{k+1}\mathbf{p}_{k+1}\mathbf{e}_{k+1}^\top, \tag{9}$$

$$\mathbf{A}\mathbf{P}_k = \mathbf{Q}_k\mathbf{L}_k, \tag{10}$$

where $\mathbf{Q}_k \in \mathbb{R}^{n \times (k+1)}$ and $\mathbf{P}_k \in \mathbb{R}^{m \times k}$ have orthonormal columns, $\mathbf{L}_k \in \mathbb{R}^{(k+1) \times k}$ is a lower bidiagonal matrix, $\mathbf{e}_{k+1} \in \mathbb{R}^{k+1}$ is the $(k+1)$th standard basis vector, and $\gamma_{k+1}$ and $\mathbf{p}_{k+1}$ will be the $(k+1)$th diagonal entry of $\mathbf{L}_{k+1}$ and the $(k+1)$th column of $\mathbf{P}_{k+1}$, respectively.

Using the bidiagonalization, we derive the projection of the regularized problem (7) as follows

$$\min_{\mathbf{x} \in \mathcal{K}_k} \frac{1}{2m}\|\mathbf{A}\mathbf{x} - \mathbf{b}\|_2^2 + \frac{\alpha^2}{2}\|\mathbf{x}\|_2^2 = \min_{\mathbf{f} \in \mathbb{R}^k} \frac{1}{2m}\|\mathbf{A}\mathbf{P}_k\mathbf{f} - \mathbf{b}\|_2^2 + \frac{\alpha^2}{2}\|\mathbf{f}\|_2^2, \tag{11}$$

where we used that the columns of $\mathbf{P}_k$ form an orthonormal basis of $\mathcal{K}_k$, which also implies that $\|\mathbf{P}_k\mathbf{f}\|_2 = \|\mathbf{f}\|_2$. Next, using (10) gives

$$= \min_{\mathbf{f} \in \mathbb{R}^k} \frac{1}{2m}\|\mathbf{Q}_k\mathbf{L}_k\mathbf{f} - \mathbf{b}\|_2^2 + \frac{\alpha^2}{2}\|\mathbf{f}\|_2^2. \tag{12}$$

Using the orthonormality of the columns of $\mathbf{Q}_k$ and the fact that $\mathbf{Q}_k$ contains $\frac{\mathbf{b}}{\|\mathbf{b}\|_2}$ in its first column, we obtain the projected problem

$$= \min_{\mathbf{f} \in \mathbb{R}^k} \frac{1}{2m}\|\mathbf{L}_k\mathbf{f} - \beta\mathbf{e}_1\|_2^2 + \frac{\alpha^2}{2}\|\mathbf{f}\|^2, \tag{13}$$

where $\beta = \|\mathbf{b}\|_2$ and $\mathbf{e}_1 \in \mathbb{R}^{k+1}$ is the first standard basis vector. Here, the $k$-dimensional projected problem (13) is greatly reduced in size compared to the original $n$-dimensional problem (7). The $k$th iteration of LSQR is obtained from the solution to the projected problem and can be computed using the regularized pseudoinverse $\mathbf{L}_{k,\alpha}^\dagger$ via

$$\mathbf{P}_k\mathbf{f}_\alpha = \beta\mathbf{P}_k\mathbf{L}_{k,\alpha}^\dagger\mathbf{e}_1 \quad \text{with} \quad \mathbf{L}_{k,\alpha}^\dagger = \frac{1}{n}\left(\frac{1}{n}\mathbf{L}_k^\top\mathbf{L}_k + \alpha^2\mathbf{I}\right)^{-1}\mathbf{L}_k^\top. \tag{14}$$

**Automatic Weight Decay** In weight decay, the choice of the hyperparameter $\alpha$ is crucial in order to obtain an effective regularizing effect. One straightforward way to adaptively choose it is to perform cross-validation. Having the small-dimensional projected problem (13) allows us to test multiple candidate $\alpha$'s for cross-validation efficiently. Here, the parameter selection can be done even more effectively by using statistical criteria such as generalized cross-validation (GCV) (Golub et al., 1979; Engl et al., 1996; Hansen, 1998; Vogel, 2002), weighted GCV (Chung et al., 2008), L-Curve (Calvetti et al., 1999) and discrepancy principle (Vogel, 2002). In this paper, we use GCV for simplicity.

The idea of GCV is to pick a weight decay hyperparameter that gives good generalization power. Specifically, it performs $m$-fold (leave-one-out) cross-validation (Kohavi et al., 1995) without solving the problem $m$ times by minimizing a loss function on the training data. Thus, it does not require validation data and is done highly efficiently using the low-rank projected solution (14).

In particular, in each iteration of the hybrid scheme, we minimize the GCV function for the projected problem (13) given by

$$G_{\mathbf{L}_k, \beta \mathbf{e}_1}(\alpha) = \frac{k \|(\mathbf{I} - \mathbf{L}_k \mathbf{L}_{k,\alpha}^\dagger) \beta \mathbf{e}_1\|_2^2}{(\operatorname{trace}(\mathbf{I} - \mathbf{L}_k \mathbf{L}_{k,\alpha}^\dagger))^2}. \tag{15}$$

Here, the SVD of $\mathbf{L}_k$ is performed quickly because it is small in size (($k{+}1$)-by-$k$). We can then plug the SVD into (15). The minimization will become a simple one-dimensional problem and can be done by standard algorithms. This renders the GCV minimization effective, which needs to be done in each iteration.

In principle, we can compute the GCV also for the full problem (7) to obtain an estimation for $\alpha$. This is a classical approach to implement LSQR, as a generic implementation of LSQR requires $\alpha$ to be pre-specified and fixed throughout the iterations. However, this would be computationally very expensive because the full SVD of $\mathbf{A}$ is required. For more details, see Chung et al. (2008); Chung & Gazzola (2024).

**Advantages of the Hybrid Method**  The regularization imposed by the hybrid method provides important distinct advantages over weight decay and early stopping. First, in (15), the hybrid method performs an adaptive weight decay by dynamically selecting $\alpha$ efficiently using information from the small (but increasing) dimension Krylov subspace. Specifically, the hybrid method chooses $\alpha$ in (8) based on the singular values of the projected problem (15) and these singular values are increasingly better approximations of the full dimension problem (7). Moreover, this hyperparameter selection does not require a dedicated validation dataset. Secondly, one can also use the GCV function value as a criteria for early stopping, see Chung et al. (2008); Björck et al. (1994). This effectively employs a safeguard regularization. Moreover, the automatic weight decay imposed by the hybrid method ensures that the computed solutions are much less sensitive to the precise stopping iteration; see the comparison between Figure 2 and Figure 1(a)-(b). The insensitivity allows the usage of fast converging iterative methods. This combination of automatic tuning of weight decay hyperparameters, safeguarded regularization, and automatic iteration stopping criteria is very powerful; see experimental results in Section 5 and Appendix A.

## 4  Ill-posedness of Random Feature Model Training

In this section, we describe the experimental setup for random feature models (RFM). Using an inverse problems perspective, we then draw a connection between the model width, its generalizability, and the ill-posedness of its training problem. This connection motivates the use of regularization and will then be used to set up the numerical experiments in Section 5 and Appendix A.

**Experimental Setup**  We consider a supervised learning problem arising in the training of RFM. Given the matrix of input features $\mathbf{Y} \in \mathbb{R}^{m \times n_f}$ and the matrix of corresponding outputs $\mathbf{B} \in \mathbb{R}^{m \times n_c}$. Here, $n_f$ is the number of input features, $n_c$ is the number of output features (e.g., the number of classes), and the $m$ examples are stored row-wise. The idea in RFM is to transform the input features by applying a random nonlinear transformation $f : \mathbb{R}^{n_f} \to \mathbb{R}^n$ to each row in $\mathbf{Y}$ and then train a linear model to approximate the relationship between $f(\mathbf{Y})$ and $\mathbf{C}$. Here, the transformation $f(\mathbf{Y})$ is applied row-wise and yields a new representation of the features in $\mathbb{R}^{m \times n}$. The dimension $n$ controls the expressiveness of the RFM and can be chosen arbitrarily; generally, larger values of $n$ increase the expressiveness of the RFM.

~~Similar to Ma et al. (2020) we define our RFM using a randomly generated matrix $\mathbf{K} \in \mathbb{R}^{n_f \times (n-1)}$, a bias vector $\mathbf{d} \in \mathbb{R}^{n-1}$, and an activation function $a : \mathbb{R} \to \mathbb{R}$, as~~ The general formulation of an RFM is given by

$$\mathbf{A} = f(\mathbf{Y}) = \begin{bmatrix} a(\mathbf{YK} + \mathbf{1}_m \mathbf{d}^\top) & \mathbf{1}_m \end{bmatrix}, \tag{16}$$

where the activation function $a$ is applied element-wise, and $\mathbf{1}_m \in \mathbb{R}^m$ is a vector of all ones used to model a bias term. ~~In our experiments, we use the ReLU activation function $a(x) = \max(x, 0)$ and~~

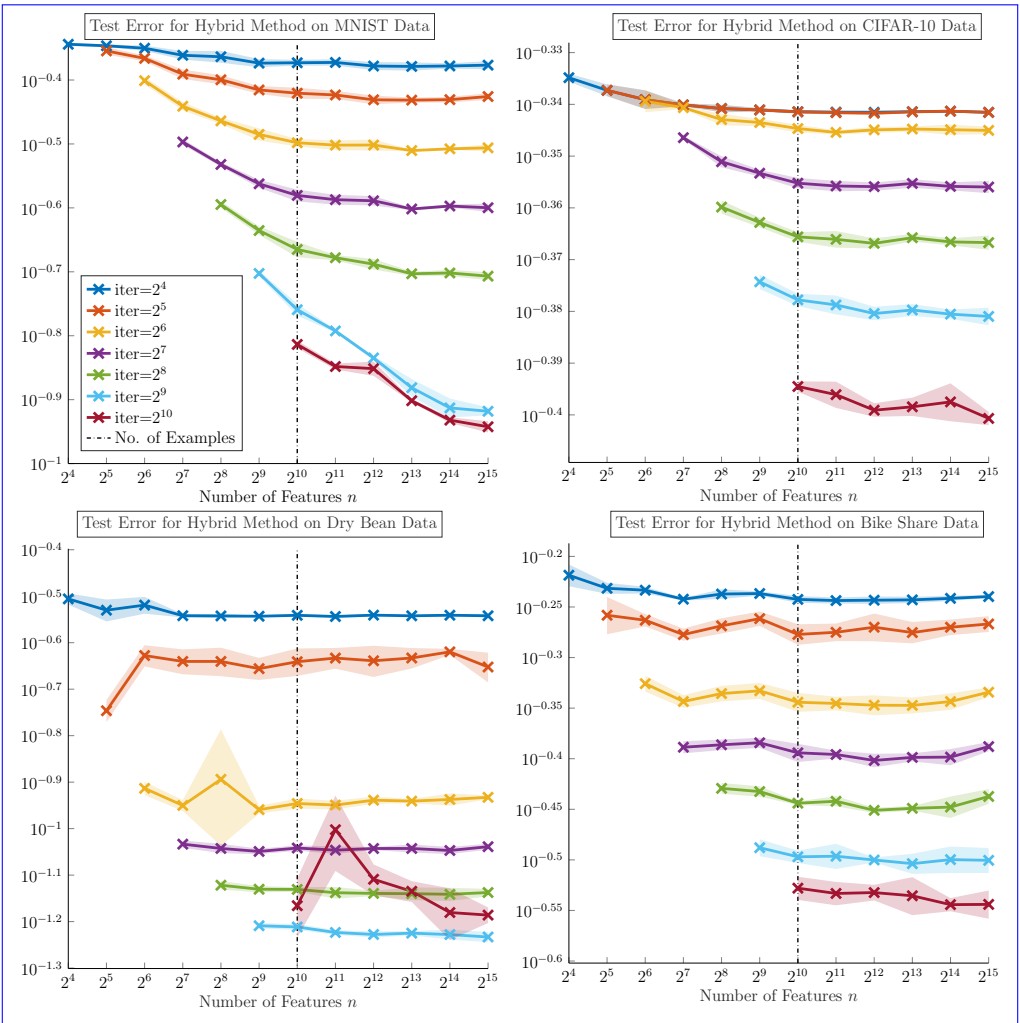

Figure 2: The results obtained by the hybrid method with different numbers of iterations on ~~MNIST and CIFAR10 data~~four datasets. The results when the iteration number is greater than $n$ are not shown as the algorithm converges after $n$ iterations. This is because the Krylov subspace is $\mathbb{R}^n$, and the projected problem becomes the original problem. The curves represent the mean over 5 random trials, while the shaded regions indicate the range spanning $\pm 1$ standard deviation from the mean. We see that the hybrid method does not exhibit semiconvergence, and the resulting test errors are not sensitive to the stopping criteria.

~~generate $\mathbf{K}$ and $\mathbf{d}$~~ We experiment with two setups for RFM. In the first setup, we generate $\mathbf{K} \in \mathbb{R}^{n_f \times (n-1)}$ and $\mathbf{d} \in \mathbb{R}^{n-1}$ using a uniformly random distribution drawn from a unit sphere and use the ReLU activation function $a(x) = \max(x, 0)$; see also Ma et al. (2020). In the second setup, we generate $\mathbf{K} \in \mathbb{R}^{n_f \times (n-1)/2}$ using a standard Gaussian distribution, set $\mathbf{d}$ to be a zero vector, and define the activation $a : \mathbb{R} \to \mathbb{R}^2$ as $a(x) = [\cos(x)\ \sin(x)]$. This setup is the random Fourier features with a Gaussian kernel in Rahimi & Recht (2007). Due to space constraints, we present the figures for the first setup in the main context and the figures for the second setup in Appendix A. We note that our findings are consistent across the two different experimental setups.

The RFM training consists of finding the linear transformation $\mathbf{X} \in \mathbb{R}^{n \times n_c}$ such that $\mathbf{AX} \approx \mathbf{B}$. As in Ma et al. (2020), we measure the quality of the model using the least squares loss function and consider the unregularized regression problem, i.e.,

$$\min_{\mathbf{X} \in \mathbb{R}^{m \times n_c}} \frac{1}{2m} \|\mathbf{AX} - \mathbf{B}\|_F^2, \tag{17}$$

where $\| \cdot \|_{\mathrm{F}}$ is the Frobenius norm. Problem (17) is separable, which means that it can be decoupled into $n_{\mathrm{c}}$ least-squares problems each of which determines one column of the solution. Therefore, without loss of generality, we focus our discussion on the case $n_{\mathrm{c}} = 1$. In this case, the training problem (17) becomes (1), with $\mathbf{b} \in \mathbb{R}^n$ being the output labels. For example, when one chooses $\mathbf{b}$ as the $i$th column of $\mathbf{B}$, (1) gives the $i$th column of the solution to (17).

**Double Descent and Ill-posedness**  The key hyperparameter of an RFM is the dimensionality of the feature space, $n$. For a given number of training data $m$, it has been observed (Belkin et al., 2019) that the RFM's generalization gap has three stages of behavior when $n < m$, $n = m$, and $n > m$, respectively, see Figure 3(a)-(~~b)~~ d) and Figure 7(a)-(d). This behavior is called the double descent phenomenon in the literature and can be explained using a data-fitting viewpoint (Belkin et al., 2019).

- When $n < m$, the learning problem (1) is overdetermined. That is, there are more equations than variables in $\mathbf{Ax} = \mathbf{b}$. There is no solution to perfectly describe the input-output relation in general. Hence as $n$ increases, we can fit both the training and test data better and the generalization gap decreases.

- When $n = m$, $\mathbf{A}$ is square and, in our experience, invertible. Thus the optimal solution to (1) is unique and satisfies $\mathbf{Ax} = \mathbf{b}$. In order words, the training data can be fitted perfectly, and the training loss is essentially zero. However, the uniqueness of the optimal $\mathbf{x}$ also implies that we have no choice but to perfectly fit to the weakly present features in $\mathbf{A}$, which are not relevant to the classification. The perfect loss on the training data combined with an increase in test loss then causes a spike in the generalization gap.

- When we further increase $n$ such that $n > m$, the problem is underdetermined, and there are infinitely many optimal $\mathbf{x}$ to achieve an objective function value of zero. It has been observed that selecting the solution with the minimal norm reduces the risk of fitting weakly present features; see, e.g., Belkin et al. (2019). Therefore, generally, the generalization gap decreases as $n$ grows.

In this work, we use an inverse problems perspective to relate this behavior in the generalization gap to the ill-posedness of the training problem. We illustrate the quantities in (2) for the three stages of the double descent in Figures 3(~~e~~e)-(~~e)~~ g) and 7(e)-(g) using the CIFAR10 example. Here, we plot $\sigma_j$, $|\mathbf{u}_j^\top \mathbf{b}|$ and $|\mathbf{u}_j^\top \mathbf{b}|/\sigma_j$ ~~in Figures 3(c)-(e)~~; the resulting plot is known as a Picard plot (Hansen, 2010). From the decay of the singular values (see the blue line), we see that $n = m$ leads to an ill-posed problem as the $\sigma_j$ decays to zero with no significant gap. Also, for $n = m$, the magnitude of $\mathbf{u}_j^\top \mathbf{b}$ (see red line) remains approximately constant. This combination causes a surge of $|\mathbf{u}_j^\top \mathbf{b}|/\sigma_j$ (see yellow line), which causes an increase of the norm of $\mathbf{w}$; see the red dashed line in Figure 3(a)-(~~b)~~ d) and Figure 7(a)-(d). When $n \neq m$ the problem is not ill-posed as the decay of the singular values is less pronounced. This relationship between RFM's deteriorated generalizability, its model width, and its ill-posedness allows us to create training problems of different levels of ill-posedness in our experiments by simply varying the model width.

## 5  Numerical Experiments

In this section, we compare and discuss the numerical results achieved with different regularization schemes on problems of different levels of ill-posedness.

**~~Examples from Image Classification~~Datasets**  In our experiments, we use ~~two common image classification~~ a total of four common benchmarks to illustrate the techniques. ~~Specifically, we use~~ We use two widely used image classification datasets: 1. the MNIST dataset~~consisting~~, which consists of $28 \times 28$ gray-scale images of hand-written digits (LeCun et al., 1990) and 2. the CIFAR 10 dataset (Krizhevsky et al., 2009)~~consisting~~, which consists of $32 \times 32$ RGB images of objects divided into one of ten categories. We also use two common dataset from the UC Irvine Machine Learning Repository: 3. Dry Bean classification dataset, which contains 16 morphological features and 7 classes representing different types of dry beans and 4. Bike sharing regression dataset, which provides 10 input features (including weather conditions and date

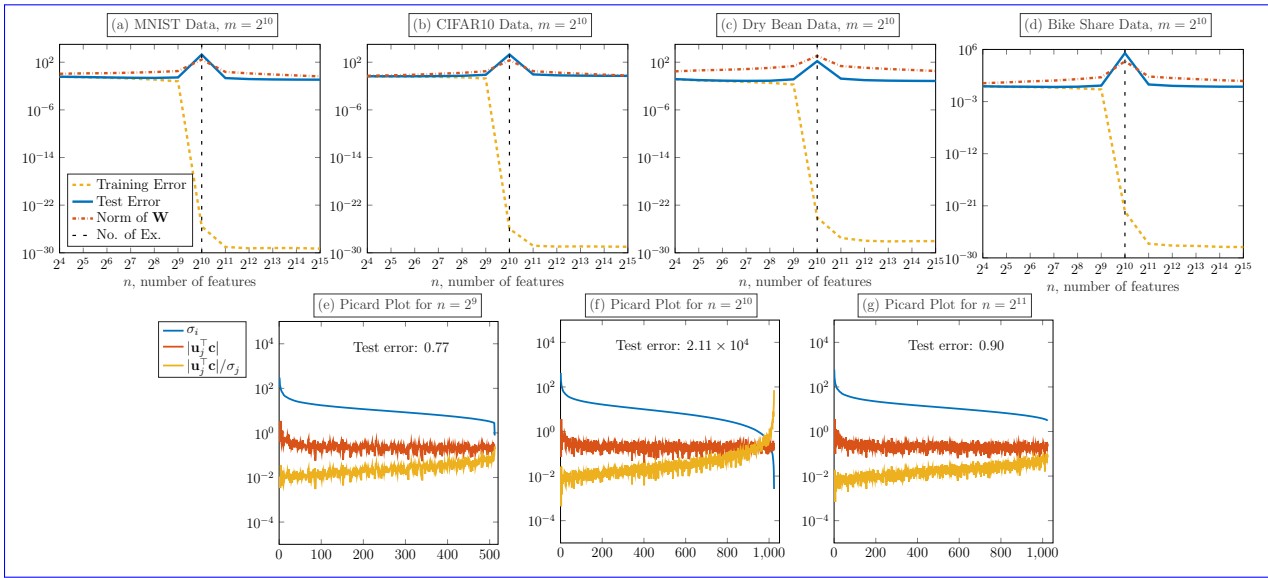

Figure 3: (a)-(~~b~~d): The deterioration of generalization observed in random feature model on MNIST~~and~~, CIFAR10, Dry Bean and Bike Share data when $n = m$. This is also called the double descent phenomenon in the literature. (~~e~~e)-(~~e~~g): The Picard plot for $\mathbf{A}$ and $\mathbf{b}$ on CIFAR10 data with $m = 1024$ and $n = 512$ (overdetermined), 1024 (unique) and 2048 (underdetermined). All the values are averaged over 5 random trials. The top, middle and bottom curves are $\sigma_j$, $|\mathbf{u}_j^\top \mathbf{c}|$ and $|\mathbf{u}_j^\top \mathbf{c}|/\sigma_j$ defined in (2), respectively. When $n = m$, there is a smooth plummet in $\sigma_j$ at the end, and it renders $|\mathbf{u}_j^\top \mathbf{c}|/\sigma_j$ large and the training problem ill-posed. These large values dominate the optimal solution $\mathbf{X}$. Thus, there is a spike in the norm of $\mathbf{X}$ and hence the testing loss.

information) and 2 target outputs (the count of casual and registered rental bikes). From each dataset, we randomly sample $m = 1,024$ training images and their labels. Each dataset also contains $m_{\text{test}} = 10,000$ labeled test images, which we use to compute the generalization gap of the trained model.

**Baseline** We compare the hybrid scheme with early stopping and weight decay regularization. We also include the unregularized model (the solution to (17)) as a basic result. To obtain competitive baseline results, we optimize the weights for the two classical regularizers using the test data. It is important to emphasize that this is neither practical nor advised in realistic applications. However, our goal is to obtain competitive baselines to compare with the hybrid scheme in which neither the test data nor some validation data is used. In contrast, our hybrid method does not have access to any test data.

We optimize the weights for each of the datasets and different widths of the RFM. To this end, we compute the (economic) SVD of the feature matrix $\mathbf{A}$ and use the filter factors in (6) and (8), respectively, to efficiently compute the optimal weights for different choices of $t$ and $\alpha$. Then, we minimize the test error over the hyperparameters using the one-dimensional optimization method `fminsearch` in MATLAB. To reduce the risk of being trapped in a suboptimal local minimum, we first evaluate the test loss at 100 points spaced equally on the logarithmic axes shown in Figure 1 and initialize the optimization method at the hyperparameter with the lowest test loss.

It is important to emphasize that, in contrast to early stopping and weight decay regularization, the hyperparameter $\alpha$ of the hybrid method is automatically and efficiently chosen in each iteration. Moreover, it does not require a dedicated validation dataset. ~~We~~Since the hybrid scheme is insensitive to the stopping iteration (see Figure 2 and Figure 6), we recommend choosing the number of iterations to match the computational budget. Alternatively, since the GCV function value (15) indicates performance during cross-validation, it can also serve as a stopping criterion.

**Experimental Setup**   Using the correlation between the ill-posedness and the dimension of the random feature $n$, we generate training problems with different levels of ill-posedness by varying $n$, the width of the RFM. We then use the training problems to test the performance of the hybrid and classical regularization schemes. The training problem is ill-posed when $n$ is close to $m$ and is the most ill-posed when $n = m$. In this case, the unregularized model leads to a large generalization gap, and we can investigate the methods' ability to improve generalizability. When $n$ is far away from $m$, the problems are relatively well-posed, and the models obtained from the unregularized problem (1) have a small generalization gap. In this situation, a desirable regularization scheme should improve or obtain a similar generalization to the unregularized solution.

**Results and Discussion**   We apply the hybrid method to the ~~training problems arising from image classification. We~~ four training problems. The results are reported based on five random trials. In each trial, we generate a different random feature matrix and randomly sample the training data. We experiment with two different RFM setups; see Section 4 for more details. Due to space limit, the figures for one RFM setup are shown in Appendix A. We show the test error for an increasing number of iterations in Figure 2 ~~. For all $n$ and~~ ~~for both datasets~~ and Figure 6. Generally, the test error of the hybrid scheme decreases with the number of iterations until it reaches $n$ when the bidiagonalization is exact. The only exception is for the dry beans dataset, where the generalization gap for $k = n$ is larger than that for $n/2$, especially around the interpolation threshold. A possible remedy to avoid this overfitting is adjusting the weighting parameter in the wGCV. This is in stark contrast to the gradient flow scheme (see Figure 1 and Figure 5) for which semiconvergence is observed, and an adequate stopping rule is needed to avoid large generalization gaps.

We compare the results of different regularization schemes in Figure 4 and Figure 8. For the hybrid regularization scheme, we set the number of iterations to $\min(n, 2^{10})$; see also Figure 2 and Figure 6. The hybrid scheme achieves competitive test errors even though it does not use the test data set. Remarkably, the hybrid method's solution is nearly on par with the optimally tuned classical schemes when the problem is the most ill-posed ($n = m$). For well-posed problems (when $n$ is far away from $m$), hybrid regularization achieves near-optimal regularizing effects. It also has similar performance as the unregularized solutions for well-posed problems, and it improves the unregularized solution for the CIFAR10 experiments when $n > 2^7$. These results demonstrate the potential of the hybrid method as a generic algorithm to reliably train machine learning models. It can automatically and efficiently solve training problems of different levels of ill-posedness. It is important to note that the ill-posedness of a training problem is not known a priori, and our results show that hybrid methods can be employed for problems of different levels of ill-posedness. Specifically, for ill-posed problems, it can effectively serve as a safeguard to prevent the deterioration in the generalization gap. It can also be used for well-posed problems, as it obtains performance comparable to optimally tuned classical regularization methods and similar to the unregularized problem.

## 6   Conclusion

We demonstrated the potential of hybrid regularization in training random feature models that generalize well. Hybrid regularization is a class of effective regularization methods commonly used to tackle ill-posed linear inverse problems. It synergistically combines early stopping and weight decay. Hybrid regularization avoids the drawbacks of the two classical regularization schemes, where it does not require cumbersome hyperparameter tuning, solution of multiple instances of the learning problem, and a dedicated validation set. Hybrid regularization is computationally efficient thanks to early stopping and performing hyperparameter selection on a low-dimensional subspace.

In our numerical experiments, we considered problems arising from training random feature models. By varying the width of the models, we created training problems of different levels of ill-posedness. Our empirical findings showed that the hybrid regularization obtained competitive test errors with optimally tuned classical regularization methods in all well- and ill-posed problem instances. This suggested the prospect of hybrid regularization methods as a generic optimization algorithm to train different machine learning models that generalize well. In future works, we plan to extend hybrid methods to more general learning problems, particularly with other loss functions, machine learning models, or stochastic optimization. We provide our MATLAB codes for the numerical experiments at `Github-link-made-anonymous-for-reviewing`.

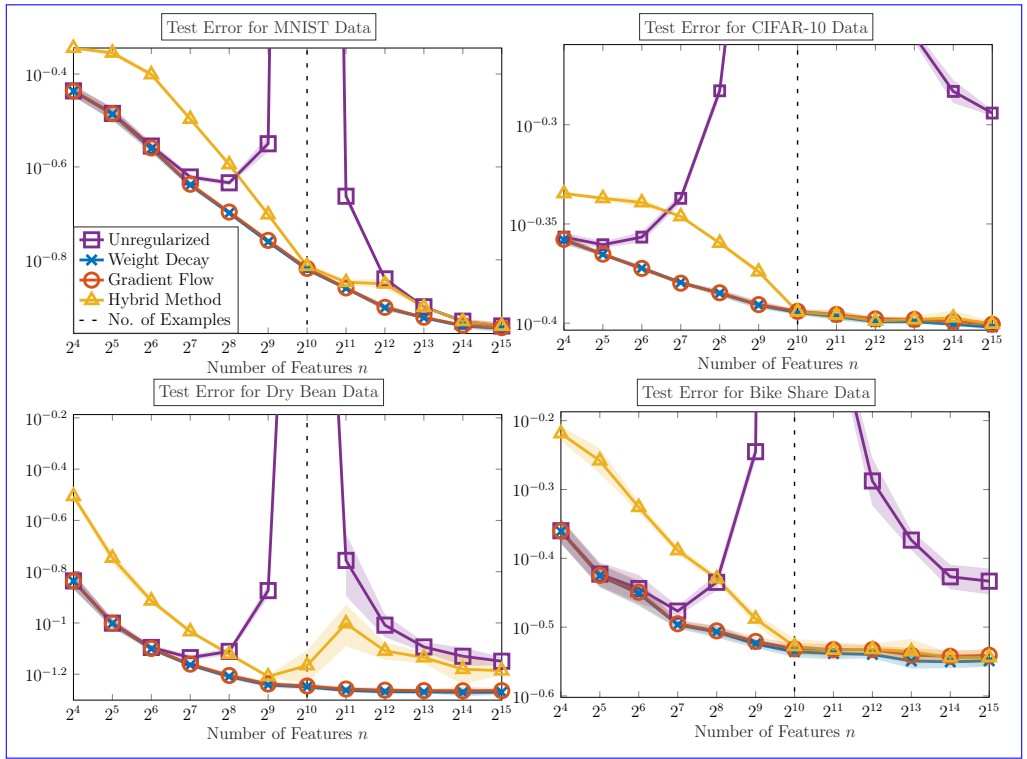

Figure 4: The results obtained by gradient flow, weight decay, and the hybrid method. For gradient flow and weight decay, the optimal testing losses over time and $\alpha$, respectively, are reported. Specifically, we minimize the testing losses with respect to the hyperparameters. For the hybrid method, we determine the weight decay hyperparameters using the training data only and report the test loss with $\min(n, 1024)$ iterations. The curves represent the mean over 5 random trials, while the shaded regions indicate the range spanning $\pm 1$ standard deviation from the mean. In the top row, the shaded regions are barely visible because the standard deviations are too small.

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

# A  Additional Experimental Results

We show the figures for the additional experimental results for the random Fourier features with a Gaussian kernel (Rahimi & Recht, 2007). We remark that the experimental results are consistent with another RFM setup reported in the main context, and the hybrid method is effective for both setups. Recall that the corresponding RFM is given by

$$\mathbf{A} = f(\mathbf{Y}) = \left[\ a(\mathbf{YK} + \mathbf{1}_m \mathbf{d}^\top)\quad \mathbf{1}_m\ \right].$$

Here, we generate $\mathbf{K} \in \mathbb{R}^{n_f \times (n-1)/2}$ using a standard Gaussian distribution, set $\mathbf{d}$ to be a zero vector, and define the activation $a : \mathbb{R} \to \mathbb{R}^2$ as $a(x) = [\cos(x)\ \sin(x)]$.

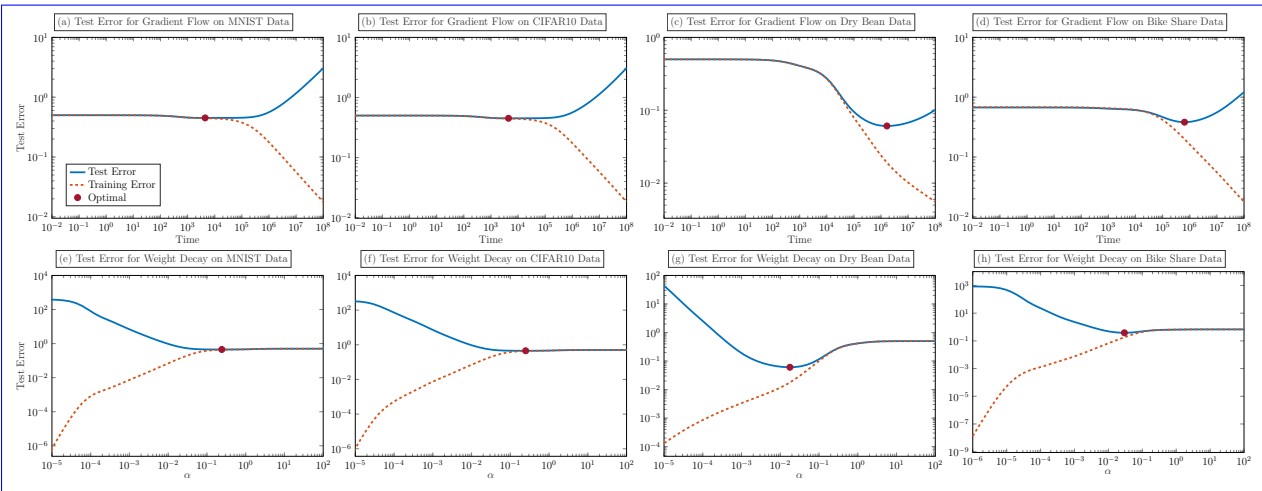

Figure 5: (a)-(d): The results of applying gradient flow (GF) (5) to four training problems at different time $t$. (e)-(h): The results obtained from weight decay (7) with different hyperparameter $\alpha$. The optimal stopping time/regularization hyperparameter is highlighted. We can see that both methods exhibit a semiconvergence behavior. In particular, their generalizability depends on the choice of hyperparameters, which varies from problem to problem and has to be made judiciously. Determining the optimal hyperparameters requires access to the test data. Yet, the access to test data is prohibited during training. Here, the features are given by the random Fourier features with a Gaussian kernel (Rahimi & Recht, 2007).

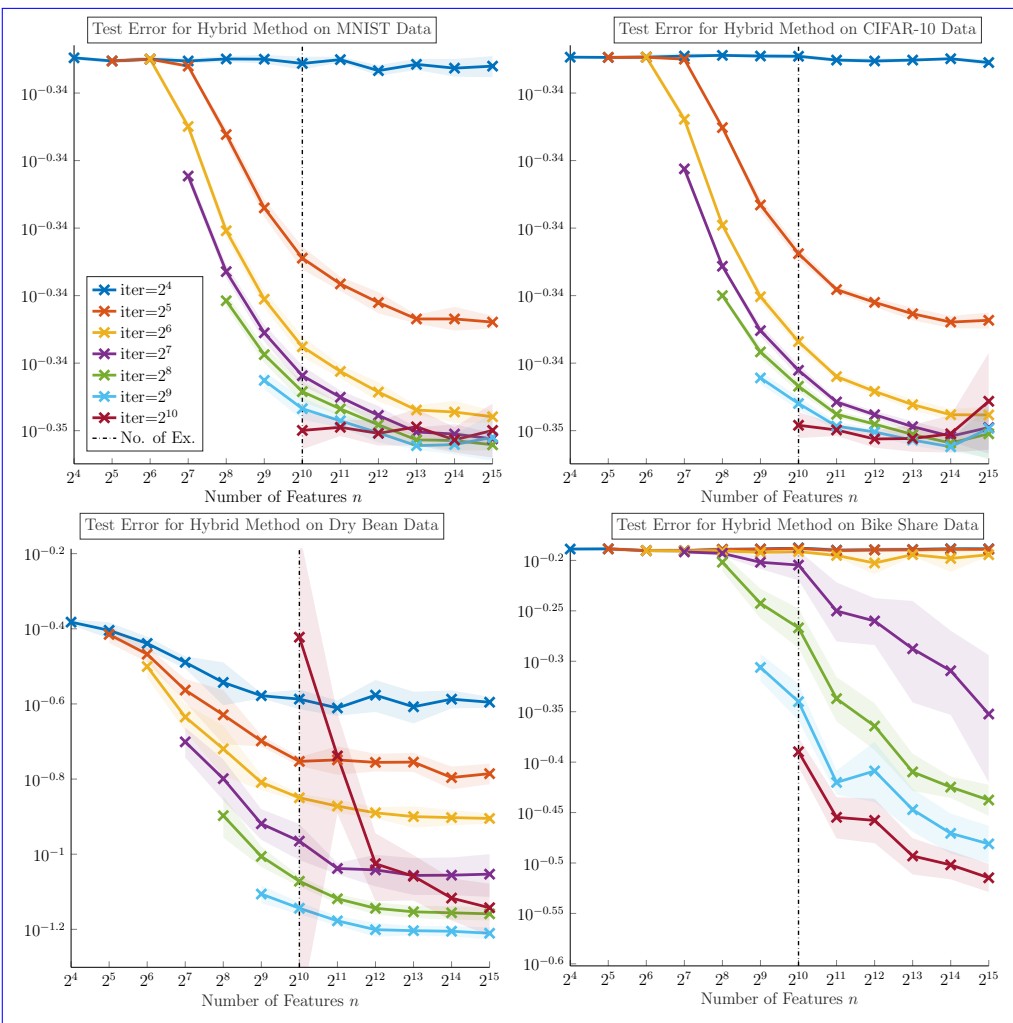

Figure 6: The results obtained by the hybrid method with different numbers of iterations on four datasets. The results when the iteration number is greater than $n$ are not shown as the algorithm converges after $n$ iterations. This is because the Krylov subspace is $\mathbb{R}^n$, and the projected problem becomes the original problem. The curves represent the mean over 5 random trials, while the shaded regions indicate the range spanning $\pm 1$ standard deviation from the mean. We see that the hybrid method does not exhibit semiconvergence, and the resulting test errors are not sensitive to the stopping criteria. Here, the features are given by the random Fourier features with a Gaussian kernel (Rahimi & Recht, 2007).

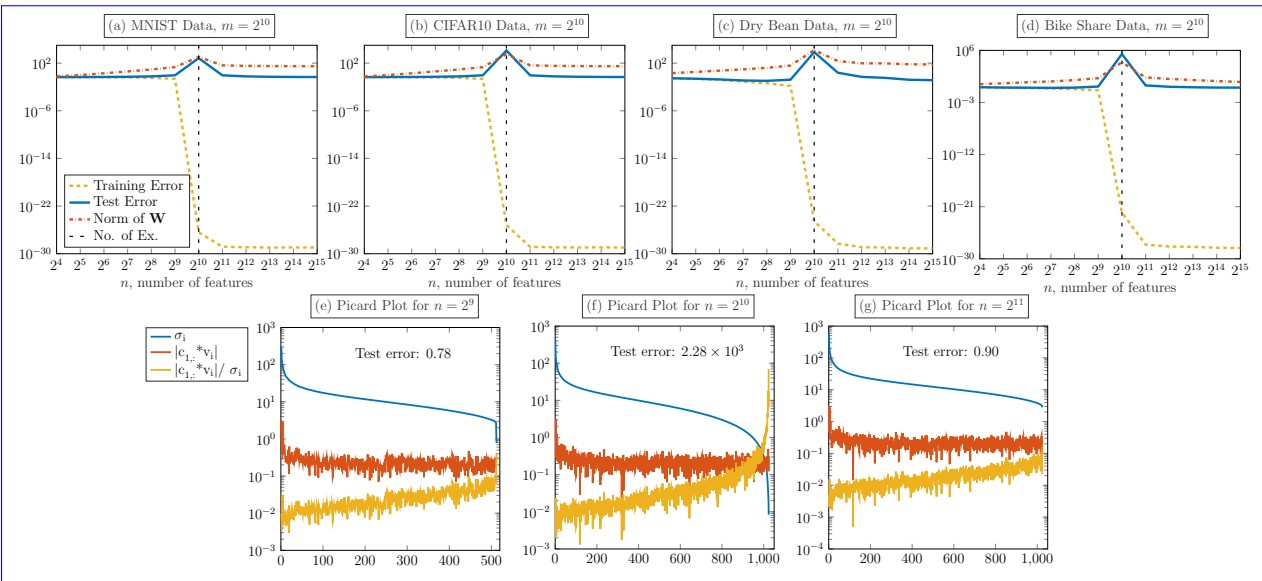

Figure 7: (a)-(d): The deterioration of generalization observed in random feature model on MNIST, CIFAR10, Dry Bean and Bike Share data when $n = m$. This is also called the double descent phenomenon in the literature. (e)-(g): The Picard plot for $\mathbf{A}$ and $\mathbf{b}$ on CIFAR10 data with $m = 1024$ and $n = 512$ (overdetermined), 1024 (unique) and 2048 (underdetermined). All the values are averaged over 5 random trials. The top, middle and bottom curves are $\sigma_j$, $|\mathbf{u}_j^\top \mathbf{c}|$ and $|\mathbf{u}_j^\top \mathbf{c}|/\sigma_j$ defined in (2), respectively. When $n = m$, there is a smooth plummet in $\sigma_j$ at the end, and it renders $|\mathbf{u}_j^\top \mathbf{c}|/\sigma_j$ large and the training problem ill-posed. These large values dominate the optimal solution $\mathbf{X}$. Thus, there is a spike in the norm of $\mathbf{X}$ and hence the testing loss. Here, the features are given by the random Fourier features with a Gaussian kernel (Rahimi & Recht, 2007).

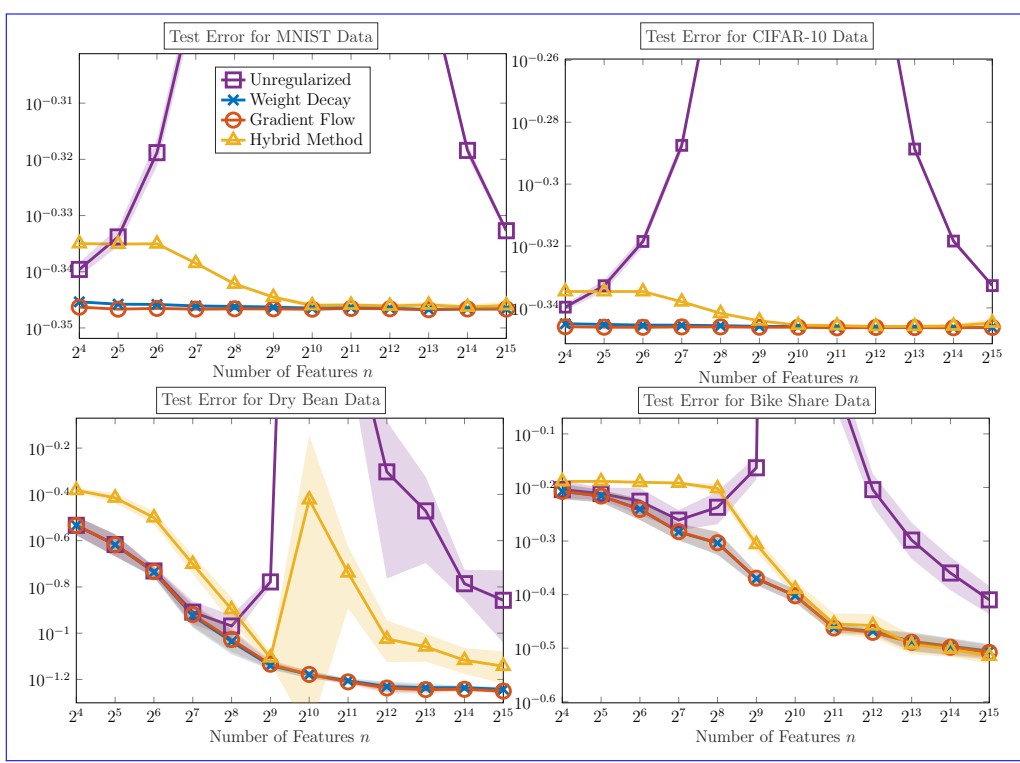

Figure 8: The results obtained by gradient flow, weight decay, and the hybrid method. For gradient flow and weight decay, the optimal testing losses over time and $\alpha$, respectively, are reported. Specifically, we minimize the testing losses with respect to the hyperparameters. For the hybrid method, we determine the weight decay hyperparameters using the training data only and report the test loss with $\min(n, 1024)$ iterations. The curves represent the mean over 5 random trials, while the shaded regions indicate the range spanning $\pm 1$ standard deviation from the mean. In the top row, the shaded regions are barely visible because the standard deviations are too small. Here, the features are given by the random Fourier features with a Gaussian kernel (Rahimi & Recht, 2007).

