# OpenReview forum: "Hybrid Regularization Methods Achieve Near-Optimal Regularization in Random Feature Models"
_TMLR — Rejected by TMLR_

### Review · Reviewer_NMyb · 2024-07-28

**Summary Of Contributions:**

This work considers the use of hybrid regularization methods to avoid overfitting and to generalize well in regression models, specifically random feature models (RFMs). Classical ways to regularize such approaches include weight decay ($\ell_2$-regularization) or early-stopped gradient descent. The challenge with these approaches, however, is that they require hyperparameter tuning to achieve good generalization, and optimal regularization essentially requires knowledge of the test data. This work shows that hybrid regularization methods can take advantage of both of these approaches' benefits while achieving strong generalization error. This method works by sequentially solving small least squares problems projected onto $k$-dimensional Krylov subspaces. The power of this approach lies in that the weight decay parameter can be found via generalized cross-validation, since the dimensionality of the problem $k$ is much smaller than the ambient dimension $n$. RFMs are chosen as a test-bed since one can naturally tune the level of overparameterization. An example on MNIST and CIFAR shows that the hybrid regularization method achieves test error nearly on par with optimally tuned early stopping and weight decay, which require knowledge of the test data.

**Audience:**

Yes

**Broader Impact Concerns:**

There are no broader impact concerns.

**Claims And Evidence:**

Yes

**Requested Changes:**

- Regarding my point above, I think that the paper would be strengthened with additional experiments analyzing how the performance of hybrid regularization changes under different RFM settings. For example, the type of random matrix $K$ and activation function $a(\cdot)$ influence the spectrum of the matrix $A$. How do changes in the RFM model influence the results? If the spectrum of $A$ changes (e.g., slower or faster decay in singular values) influence these results and the performance of hybrid regularization?
- In equation (3), $\phi_j$ should be a function of a parameter as it is in equations (4) and (6)

**Strengths And Weaknesses:**

**Strengths**:
- Overall, the paper is well-written with a clear motivation and easy to read. I also appreciate the reasoning on using random feature models as a test-case since one can easily tune the level of ill-posedness of the problem.
- The experiments on MNIST and CIFAR show in a specific setting that hybrid regularization indeed achieves small generalization error and competitive with optimally-tuned weight decay and early stopping.

**Weaknesses**:
- I think that the paper would be strengthened by a more comprehensive analysis of hybrid regularization for RFMs. In particular, the main result of the paper is a single experiment conducted on two toy datasets. While this presents nice evidence for the claim that hybrid regularization aids in achieving small generalization error, it would be interesting for a deeper investigation into when this works well and when it doesn't. Please see the requested changes below for additional experimental suggestions.

---

> ### Author Response · Authors · 2024-09-25
> **Authors' response**
>
> Q: In equation (3), $\phi_j$ should be a function of a parameter as it is in equations (4) and (6) \
> A: Thank you for pointing this out. We have added a sentence emphasizing the dependence of $\phi_j$ on the regularization hyperparameter. Since it depends on a different hyperparameter based on the regularization method (e.g., singular value $\tau$ for SVD, stopping time $t$ for early stopping, or $\alpha$ for weight decay), for simplicity of exposition, we do not show the dependency explicitly in equation (3), and we specify the dependency in the following discussion.
>
> Q: The type of random matrix and activation function influence the spectrum of the matrix. How do changes in the RFM model influence the results? If the spectrum of changes (e.g., slower or faster decay in singular values) influence these results and the performance of hybrid regularization? \
> A: Given the short timeline, we have not systematically compared different types of random matrix and activation functions, but  we have added experiments on two additional datasets. In addition, we also re-ran the existing experiments over five random trials and report the confidence intervals. At each random trial, the data are randomly split into training and test sets, and the random feature matrix (equation (16)) is randomly generated. These additional experiments provide more random matrices with different spectral behaviors to test hybrid methods. The new results are consistent with the original findings, i.e., hybrid regularization methods are robust and competitive with \textit{optimally tuned} classical regularization methods. \
>     In the meantime, we are also running addition experiments with a different random matrix and activation function. Specifically, we are using the Random Fourier Features with a Gaussian kernel from [1] and a softplus activation function. We will update the manuscript once the results are obtained.
>
> [1] Rahimi, A., & Recht, B. (2007). Random features for large-scale kernel machines. Advances in neural information processing systems, 20.

---

> ### Author Response · Authors · 2024-10-07
> **Authors' response**
>
> Based on the Reviewer NMyb's comments, we also performed additional experiments using a different random feature model (RFM). Specifically, we experimented with the random Fourier feature model in [1], where the random features are generated from a Gaussian, and the activation function is given by cos and sin functions.
>
> The additional results are reported in the Appendix. We note that using a different RFM results in a similar behavior in the spectrum of the data matrix; the problem is the most ill-posed when number of training data equals number of random features, see Figures 3 and 7.
>
> With the additional datasets and random feature model, we now have 8 experiments (four datasets and 2 models), as compared to 2 experiments in the initial submission. We remark that the new results are consistent with the original findings, and hybrid methods are effective and competitive with the comparing methods.
>
>
> [1] Rahimi, A., & Recht, B. (2007). Random features for large-scale kernel machines. Advances in neural information processing systems, 20.

---

> > ### Comment · Reviewer_NMyb · 2024-10-09
> > **Thank you**
> >
> > I thank the authors for their detailed reply and updates to the paper. I also appreciate the additional experiments on different datasets and the architecture of the random feature models. These experiments certainly improve the paper and give further evidence for the method.

---

### Review · Reviewer_FRGh · 2024-08-14

**Summary Of Contributions:**

This paper is interested in the role of regularization to prevent overfitting. Specifically, the paper reasons about two common regularizers - early stopping and $l_2$ or ridge regularization. The paper argues that these methods by themselves are not sufficient for ill posed or very boisy problems. Instead they propose a new hybrid regularizer and combines the two.

The provide the description of the regularize for least squares regression. Additionally, they numerically provide examples showing that their regularizer mitigates double descent and has improved performance.

**Audience:**

Yes

**Broader Impact Concerns:**

The work doesn't require a border impact statement.

**Claims And Evidence:**

Yes

**Requested Changes:**

I am currently voting no to claims and audience. I would request the authors do the following.

1) Improve the literature review and connection to prior work. Proper contextualization is important for an audience to be interested.

2) Additionally, for the claim that the new regulaizer is superior, the authors need more evidence. Either

a) theoretically compare the risk of the solution obtained by the new method with the risk for other prior theory work

or

b) Increase the number of datasets for more convincing experimental evidence.

**Strengths And Weaknesses:**

**Strengths**

The paper is mostly clear, and the new method seems interesting, with good numerical results.

**Weaknesses**

The paper, however, has a few weaknesses. I think addressing these weaknesses would significantly improve the paper.

1) If I understand correctly, the new regularizer already exists in prior work and is not a contribution to this paper. Additionally, the regularization procedure is not clear to me. For each $k$, we get a solution to Equation (13) with $\alpha$ chosen according to Equation (15), right? How do we pick which $k$ we use? In particular, does the $k$th iterate even depend on the $k-1$st iterate? Is it not just the solution to Equation (13)?

2) Since the paper doesn't present a new method but only explores the efficacy of an existing method, its literature review and placement of this experiment in context are not sufficient.

    a. First, the paper doesn't compare against the obvious combination—that is, the early stopped ridge regularized problem. In fact, numerically, the paper doesn't seem to compare against early stopping at all.

    b. *Optimal Ridge Regularization*. Significant work has been done to understand the optimal ridge regularization. Such work should be discussed [1,2,3,4,5]

    c.* Optimal Stopping Time*. Significant work has also been done to understand the optimal stopping time. Such work should be discussed [6,7,8,9,10]. In fact, the recent work of [10] draws connections between the two.

    d. *Tikhonov Regularizers*. The paper briefly discusses Tikhonov Regularization as weight decay. However, Tikhonov Regularization can be more general [11]. Additionally, work has been done to study the risk for such regularizers for least square problems and random feature models [12, 13, 14].

3) Since the paper doesn't present a new method, the experimental evidence is weak. That is, it is only for two datasets.


[1] Nakkiran, P., Venkat, P., Kakade, S., & Ma, T. (2020). Optimal regularization can mitigate double descent. arXiv preprint arXiv:2003.01897.

[2] Wu, D., & Xu, J. (2020). On the Optimal Weighted $\ell_2 $ Regularization in Overparameterized Linear Regression. Advances in Neural Information Processing Systems, 33, 10112-10123.

[3] Hastie, T., Montanari, A., Rosset, S., & Tibshirani, R. J. (2022). Surprises in high-dimensional ridgeless least squares interpolation. Annals of statistics, 50(2), 949.

[4] Sonthalia, R., Li, X., & Gu, B. (2023). Under-parameterized double descent for ridge regularized least squares denoising of data on a line. arXiv preprint arXiv:2305.14689.

[5] Dmitry Kobak, Jonathan Lomond, and Benoit Sanchez. The Optimal Ridge Penalty for Real- World High-Dimensional Data Can Be Zero or Negative Due to the Implicit Ridge Regulariza- tion. Journal of Machine Learning Research, 2022

[6] Madhu S. Advani, Andrew M. Saxe, and Haim Sompolinsky. High-dimensional dynamics of generalization error in neural networks. Neural Networks, 132:428–446, 2020. DOI: 10.1016/ j.neunet.2020.08.022. arXiv: 1710.03667 [stat.ML]

[7] Alnur Ali, J Zico Kolter, and Ryan J Tibshirani. A continuous-time view of early stopping for least squares regression. In The 22nd international conference on artificial intelligence and statistics, pages 1370–1378. PMLR, 2019

[8] Ruoqi Shen, Liyao Gao, and Yi-An Ma. On optimal early stopping: over-informative versus under-informative parametrization. arXiv preprint arXiv:2202.09885, 2022

[9] Garvesh Raskutti, Martin J Wainwright, and Bin Yu. Early stopping and non-parametric regression: an optimal data-dependent stopping rule. arXiv preprint arXiv:1306.3574, 2013

[10] Sonthalia, R., Lok, J., & Rebrova, E. (2024). On Regularization via Early Stopping for Least Squares Regression. arXiv preprint arXiv:2406.04425.

[11] C. M. Bishop, "Training with Noise is Equivalent to Tikhonov Regularization," in Neural Computation, vol. 7, no. 1, pp. 108-116, Jan. 1995, doi: 10.1162/neco.1995.7.1.108.

[12] Dhifallah, O., & Lu, Y. (2021, July). On the inherent regularization effects of noise injection during training. In International Conference on Machine Learning (pp. 2665-2675). PMLR.

[12] Rishi Sonthalia and Raj Rao Nadakuditi. Training data size induced double descent for denoising feedforward neural networks and the role of training noise. Transactions on Machine Learning Research, 2023

[13] Li, Z., Su, W., & Sejdinovic, D. (2020). Benign overfitting and noisy features. arXiv preprint arXiv:2008.02901.

[14] Chinmaya Kausik, Kashvi Srivastava, and Rishi Sonthalia. Double Descent and Overfitting under Noisy Inputs and Distribution Shift for Linear Denoisers. Transactions on Machine Learning Research, 2024

---

> ### Author Response · Authors · 2024-09-25
> **Authors' response**
>
> Q: If I understand correctly, the new regularizer already exists in prior work and is not a contribution to this paper. \
> A: Yes, the new regularizer is known to be effective in the inverse problem literature. However, this method is relatively unknown in the machine learning community. We illustrate that the method achieves competitive performance on problems with different levels of ill-posedness, when compared with optimally tuned classical regularizers, while avoiding their drawbacks.
>
> Q: For each $k$, we get a solution to Equation (13) with $\alpha$ chosen according to Equation (15), right? \
> A: Correct. We select the $\alpha$ that minimizes (15). Note that this problem is a one-dimensional optimization problem and that the objective function only involves matrices of size $(k+1)$ by $k$.
>
> Q: In particular, does the $k$th iterate even depend on the $k-1$st iterate? Is it not just the solution to Equation (13)? \
> A: At each LSQR iteration, we expand the matrix $L_k$ (equation (9)) using the results from the previous iteration. Specifically, at the $k$th iteration, we obtain $L_k \in \mathbb{R}^{(k+1) \times k}$ by appending new entries into $L_{k-1} \in \mathbb{R}^{k \times (k-1)}$, which was from the previous iteration. Then, we compute the $k$th iterate using (14). \
>     We note that although the $k$th iterate does not directly depends on the $k-1$st iterate, we can compute the $k$th iterate using (14) efficiently. This is because the performance is robust with respect to the stopping iteration when a hybrid regularization scheme is used; see Figure 2. So we can stop early (at iteration $k$ which is smaller than the problem size $n$) to reduce computational cost without compromising test accuracy.
>
> Q: The paper doesn't compare against the obvious combination—that is, the early stopped ridge regularized problem. In fact, numerically, the paper doesn't seem to compare against early stopping at all. \
> A: In Figure 4, we compared against early stopping and ridge regression, respectively, \textit{at their optimal performance}. Specifically, the experiments show that hybrid regularization is competitive with optimally-stopped gradient flow and ridge regression with optimally chosen parameter. In other words, the stopping time/ridge hyperparameter is chosen to minimize the test error. We note that accessing test data is impractical and only done to obtain a tough benchmark. For hybrid regularization, we neither use the test data nor need a dedicated validation dataset to select its hyperparameter using (15).  In our numerical experiments on four datasets and in the well-posed and ill-posed regime, the hybrid strategy with its standard parameters achieves near-optimal generalization.  \
>     We note that early stopped ridge regression requires hyperparameter tuning, which requires solving the \textit{original large-scale} problem many times and/or a dedicated validation dataset, reducing the amount of training data. While hybrid methods automatically choose the ridge hyperparameter using (15) efficiently and is robust with respect to the precise stopping time; see Figure 2. Moreover, (15) can also serve as a criteria for terminating the iteration \tr{(citation Golub GCV and Julianne WGCV)}. In addition, hybrid methods do not require any validation data, since minimizing (15) is equivalent to performing leave-one-out cross-validation simultaneously in a single training run. Overall, hybrid methods can be thought of as an instance of early stopped ridge regression. But hybrid methods use iteration schemes that are superior to gradient descent in terms of converging rate and select the ridge hyperparameter automatically and efficiently.
>
> Q: Literature review on optimal Ridge regularization, stopping time, and Tikonov regularization. \
> A: We have included a literature review on the suggested areas.
>
> Q: Since the paper doesn't present a new method, the experimental evidence is weak. That is, it is only for two datasets. \
> A: We have added experiments on two additional datasets: the Dry Bean and Bike Share datasets, which are among the most commonly used datasets from the UC Irvine Machine Learning Repository. And the new experimental results are consistent with our findings.

---

> > ### Comment · Reviewer_FRGh · 2024-10-02
> > **Thanks**
> >
> > I thank the authors for their comments and for answering my question. The paper does a better job of fitting in with the literature, so I will update my answer to the audience question to a yes.
> >
> > In terms of claims an evidence, I thank the authors for adding two more datasets. I am willing to update the answer to yes, but I strongly recommend the authors add more experimental evidence. Looking at the other reviews, the number of datasets seems to be a concern for the other reviewers as well

---

> ### Author Response · Authors · 2024-10-07
> **More additional experiments**
>
> Based on the Reviewer NMyb's comments, we performed additional experiments using a different random feature model (RFM). Specifically, we experimented with the random Fourier feature model in [1], where the random features are generated from a Gaussian, and the activation function is given by cos and sin functions.
>
> The additional results are reported in the Appendix. We note that using a different RFM results in a similar behavior in the spectrum of the data matrix; the problem is the most ill-posed when number of training data equals number of random features, see Figures 3 and 7.
>
> With the additional datasets and random feature model, we now have 8 experiments (four datasets and 2 models), as compared to 2 experiments in the initial submission. We remark that the new results are consistent with the original findings, and hybrid methods are effective and competitive with the comparing methods.
>
>
> [1] Rahimi, A., & Recht, B. (2007). Random features for large-scale kernel machines. Advances in neural information processing systems, 20.

---

> ### Comment · Reviewer_FRGh · 2024-10-07
> **Thanks**
>
> Thank you for adding more experiments. I think this improves the paper by providing more evidence for the method.

---

> > ### Author Response · Authors · 2024-10-08
> > **Thank you reviewer**
> >
> > We would like to thank Reviewer FRGh for their prompt reply and positive feedback.

---

### Review · Reviewer_55VL · 2024-09-12

**Summary Of Contributions:**

Regularization is a popular concept encompassing a large class of techniques to improve the ability of trained machine learning models to generalize well on unseen data as well as to find a good solution to ill-posed inverse problems.

In this paper, the authors consider hybrid regularization methods (i.e., combining several regularization methods such as early stopping and weight decay in their case) to train random feature models. A random feature model is a linear model fitted over a set of features generated by a random feature extractor.

They consider the algorithm IRhybrid_lsqr from the open source MATLAB package (Gazzola et al., 2018) that is based on the LSQR algorithm (Paige & Saunders, 1982).

The authors present the results of numerical experiments on two basic dataset, CIFAR10 and MNIST, illustrating the potential of hybrid regularization approach in training random feature models.

**Audience:**

Yes

**Claims And Evidence:**

No

**Requested Changes:**

- Weight decay/Tikhonov regularization is often referred to as “ridge penalty” in the statistical/machine learning literature. I would suggest the authors to add this terminology to appeal to a wider audience.

- The concept of generalization gap introduced by the authors page 2 seems unusual, at least for the machine learning literature. I would like the author to clarify this definition by providing references and/or sticking to a more common definition (such as, e.g., the excess risk).

- LSQR was introduced to find the least-squares solution to a large, sparse, linear system of equations but the authors never discuss the sparsity aspect. I would like the authors to comment on this point

- There are not confidence intervals appearing in the figures even though the curves that are displayed are averaged curves corresponding to repeated experiments. I would suggest the authors to add information of the variability of the plotted curves across the experiments.

- It would be more convincing if the authors could provide theoretical results illustrating the advantages of hybrid regularization methods. Moreover, the authors should extend their numerical experiments, they are limited given the empirical nature of the paper at the moment.

**Strengths And Weaknesses:**

Strengths:
- The paper is well-written, pedagogical.
- The paper could be of interest to both  the inverse problem and the machine learning communities.
- The considered approach/setting has not been considered (up to my knowledge).

Weaknesses:
- There are not theoretical results (e.g., bounds on the generalization gap).
- The empirical evidence is not sufficient. The authors only conducted experiments on two well-known datasets, MNIST and CIFAR10.
- The authors barely explain how early stopping is performed for the proposed hybrid method.

---

> ### Author Response · Authors · 2024-09-25
> **Authors' response**
>
> Q: Weight decay/Tikhonov regularization is often referred to as “ridge penalty” in the statistical/machine learning literature. I would suggest the authors to add this terminology to appeal to a wider audience. \
> A: Thank you for pointing this out. We have included this terminology and cited related literature.
>
>
> Q: The concept of generalization gap introduced by the authors page 2 seems unusual, at least for the machine learning literature. I would like the author to clarify this definition by providing references and/or sticking to a more common definition (such as, e.g., the excess risk). \
> A: We have added references to the terminology "generalization gap"
>
> Q: LSQR was introduced to find the least-squares solution to a large, sparse, linear system of equations but the authors never discuss the sparsity aspect. I would like the authors to comment on this point \
> A: This is correct, but using LSQR goes beyond sparse linear systems. LSQR is also a powerful algorithm for large-scale and dense least-squared problems. Solving the problem directly is often computationally intractable or unnecessary (e.g., because singular values decay quickly). Being an iterative method, LSQR can lead to computational savings and another benefit is that it does not require building the data matrix explicitly, but only the function handle to perform matrix-vector multiplication. This memory efficiency is especially attractive for large-scale machine learning problems arising in, e.g., image classification, where the memory requirement for storing all the features at once can be prohibitive. In these cases, LSQR can compute the approximate solution numerically more stably and with a faster convergence rate than the commonly used gradient descent. Moreover, the properties of the LSQR iterates allows us to project the problem into a low-dimensional space (13) and use (15) to automatically and efficiently select the ridge regularization hyperparameter.
>
> Q: The authors barely explain how early stopping is performed for the proposed hybrid method. \
> A: Thank you for pointing this out. We discussed this in the paragraph "Advantages of the Hybrid Method" of Section 3 and "Baseline" of Section 5. We have also added a more detailed description on this. In summary, since the hybrid scheme is insensitive to the stopping iteration (shown in Figure 2), we recommend choosing the number of iterations to match the computational budget. Alternatively, since the GCV function value (Equation 15) indicates performance during cross-validation, it can also serve as a stopping criterion.
>
> Q: There are not confidence intervals appearing in the figures even though the curves that are displayed are averaged curves corresponding to repeated experiments. I would suggest the authors to add information of the variability of the plotted curves across the experiments. \
> A: We have added $\pm$1 standard deviation, calculated from 5 random trials, to the plots reporting the numerical results.
>
> Q: It would be more convincing if the authors could provide theoretical results illustrating the advantages of hybrid regularization methods. Moreover, the authors should extend their numerical experiments, they are limited given the empirical nature of the paper at the moment. \
> A: We agree that stronger theoretical results would be a great addition, but this is beyond the scope of this paper. We have added experiments on two additional datasets: the Dry Bean and Bike Share datasets, which are among the most commonly used datasets from the UC Irvine Machine Learning Repository. And the new experimental results are consistent with our findings.

---

> > ### Comment · Reviewer_55VL · 2024-10-10
> > **Reponse**
> >
> > Thank you for your reply. I believe that there are still some important issues that should be addressed.
> >
> > - The added references for the "generalization gap" do not seem relevant. I could not find this terminology in Bengio et al., 2017 and the definition given in Johnson & Zhang, 2024 is completely different (they consider an expected error whereas whereas the generalization gap you introduce is random). The authors should change the terminology or find relevant references.
> >
> > - Regarding LSQR: I think that the authors should discuss this in the paper, otherwise the choice of LSQR does not seem reasonable/motivated enough.
> >
> > - 5 random trials seems short to estimate properly the standard deviation via Monte Carlo approximation, I would suggest the authors to run additional random trials.
> >
> > - I still regret the absence of any theoretical result, especially given that the studied method was not developed by the authors.

---

> ### Author Response · Authors · 2024-10-07
> **More additional experiments**
>
> Based on the Reviewer NMyb's comments, we performed additional experiments using a different random feature model (RFM). Specifically, we experimented with the random Fourier feature model in [1], where the random features are generated from a Gaussian, and the activation function is given by cos and sin functions.
>
> The additional results are reported in the Appendix. We note that using a different RFM results in a similar behavior in the spectrum of the data matrix; the problem is the most ill-posed when number of training data equals number of random features, see Figures 3 and 7.
>
> With the additional datasets and random feature model, we now have 8 experiments (four datasets and 2 models), as compared to 2 experiments in the initial submission. We remark that the new results are consistent with the original findings, and hybrid methods are effective and competitive with the comparing methods.
>
> [1] Rahimi, A., & Recht, B. (2007). Random features for large-scale kernel machines. Advances in neural information processing systems, 20.

---

> ### Author Response · Authors · 2024-10-11
> **Response to Reviewer**
>
> We would like to thank the reviewer for their comments.
>
> Q: The added references for the "generalization gap" do not seem relevant. I could not find this terminology in Bengio et al., 2017 and the definition given in Johnson & Zhang, 2024 is completely different (they consider an expected error whereas whereas the generalization gap you introduce is random). The authors should change the terminology or find relevant references. \
> A: Thank you for pointing this out. There was a typo in the citation, we meant Figure 5.3 of Bengio (2017), instead of Section 5.3. We removed the other citation and added two references published in respectable venues that used the same definition for generalization gap.
>
>
> Q: Regarding LSQR: I think that the authors should discuss this in the paper, otherwise the choice of LSQR does not seem reasonable/motivated enough. \
> A: Thank you for pointing this out. We added to Section 3 that LSQR is suitable for both dense and sparse large-scale problems.
>
>
> Q: 5 random trials seems short to estimate properly the standard deviation via Monte Carlo approximation, I would suggest the authors to run additional random trials. \
> A: Given the tight time constraint to run the experiments, we decided to prioritize increasing number of experiments, since it can provide more diverse settings to test out the model performance. We remark that in each trial, we ran the experiments using 12 models with different feature dimension ranging from 2^4 to 2^15. And we ran 5 trials per experiment, and with 8 experiments, we have 5×8=40 trials and 40×12=480 models in total to test the model performance. And with the additional experiments, the findings are consistent with the original ones. We also release our code so that readers can run the experiments in different settings or with more trials.
>
>
> Q: I still regret the absence of any theoretical result, especially given that the studied method was not developed by the authors. \
> A: While we agree that theoretical results would be a valuable addition, they are beyond the scope of this paper. To compensate that, we have added the additional empirical experiments, which we believe are extensive.

---

### Author Response · Authors · 2024-09-25
**Thank you reviewers**

We want to thank the reviewers for their thorough and constructive reviews. We have carefully addressed their comments and made several improvements to the paper. In the revised manuscript, the newly added contents are highlighted in blue, and the deleted contents are shown in red. We note that in page 2, there is text exceeding the right margin when using the compare package, while the original manuscript displays correctly.

We have replied to each comment to discuss our responses and changes in detail. We appreciate your time and effort in improving our work.

---

### Decision · Action_Editor_znjZ · 2024-10-29

**Recommendation:** Reject

**Comment:**

No comments beyond the "Claim and evidence" section

**Audience:**

Hybrid regularization is of great interest to the ML community.
Given the authors focus on the ML community and a ML problem, the authors could, if not to rewrite their code in python, at least provide Python wrappers for their Matlab code.

**Claims And Evidence:**

The paper studies "hybrid regularization", meaning here a combination of L2 regularization (ridge/weight decay) and early stopping when solving least squares problems.
The paper focuses on the performance of hybrid regularization for training Random Features Models (Eq 17).
It is shown to reach the same level of performance as optimally tuned L2 regularization and early stopping, while not requiring validation data nor resolution of multiple costly optimization problems for hyperparameter tuning.

Hybrid regularization has already been proposed in the literature; the paper is limited to an empirical evaluation of this technique, and does not provide theoretical insights.
Unfortunately, even after rebuttal, reviewers `FRGh` and `55VL` still found that the empirical validation was lacking. Reviewer `NMyb` also required a "deeper investigation of when the model works and does not", finding the validation "still somewhat small-scale and lacking in theory". Less importantly, Reviewer `55VL` also had legitimate claims against using "generalization gap" instead of "excess risk".

Even if the claim that using a RFM model makes the study easier, by being able to control the level of overparametrization, it seems that adding some experiments beyond the scope of RFM would show applicability in a less restrictive setting, and greatly strengthen the contribution.

**Resubmission Of Major Revision:**

The authors may consider submitting a major revision at a later time.